# WAY OFF-POLICY BATCH DEEP REINFORCEMENT LEARNING OF HUMAN PREFERENCES IN DIALOG

## ABSTRACT

Most deep reinforcement learning (RL) systems are not able to learn effectively from off-policy data, especially if they cannot explore online in the environment. This is a critical shortcoming for applying RL to real-world problems where collecting data is expensive, and models must be tested offline before being deployed to interact with the environment – *e.g.* systems that learn from human interaction. Thus, we develop a novel class of off-policy batch RL algorithms which use KL-control to penalize divergence from a pre-trained prior model of probable actions. This KL-constraint reduces extrapolation error, enabling effective offline learning, without exploration, from a fixed batch of data. We also use dropout-based uncertainty estimates to lower bound the target Q-values as a more efficient alternative to Double Q-Learning. This Way Off-Policy (WOP) algorithm is tested on both traditional RL tasks from OpenAI Gym, and on the problem of open-domain dialog generation; a challenging reinforcement learning problem with a 20,000 dimensional action space. WOP allows for the extraction of multiple different reward functions post-hoc from collected human interaction data, and can learn effectively from all of these. We test real-world generalization by deploying dialog models live to converse with humans in an open-domain setting, and demonstrate that WOP achieves significant improvements over state-of-the-art prior methods in batch deep RL.

## 1 INTRODUCTION

In order to scale deep reinforcement learning (RL) to safety-critical, real-world domains, two abilities are needed. First, since collecting real-world interaction data can be expensive and time-consuming, algorithms must be able to learn from off-policy data no matter how it was generated, or how little correlation between the data distribution and the current policy. Second, it is often necessary to carefully test a policy before deploying it to the real world; for example, to ensure its behavior is safe and appropriate for humans. Thus, the algorithm must be able to learn offline first, from a static batch of data, without the ability to explore.

This off-policy, *batch reinforcement learning* (BRL) setting represents a challenging RL problem. Most deep RL algorithms fail to learn from data that is not heavily correlated with the current policy (Fujimoto et al., 2018b). Even models based on off-policy algorithms like $Q$-learning fail to learn in the offline, batch setting, when the model is not able to explore. If the batch data is not sufficient to cover the state-action space, BRL models can suffer from *extrapolation error*, learning unrealistic value estimates of state-action pairs not contained in the batch (Fujimoto et al., 2018b). It can be impossible to correct for extrapolation error when there is a mismatch in the distribution of state-actions pairs in the batch data, and the distribution induced by the learned policy. For example, if the policy learns to select actions which are not contained in the batch, it cannot learn a reasonable value function for those actions. Figure 1 illustrates this concept, where the batch only covers a subset of possible policies. Extrapolation error is particularly problematic in high-dimensional state and action spaces (such as those inherent in language generation).

We propose to resolve these issues by leveraging a pre-trained generative model of the state-action space, $p(a|s)$, trained on known sequences of interaction data. While training with RL, we penalize divergence from this prior model with different forms of KL-control. This technique ensures that the RL model learns a policy that stays close the state-action distribution of the batch, combating

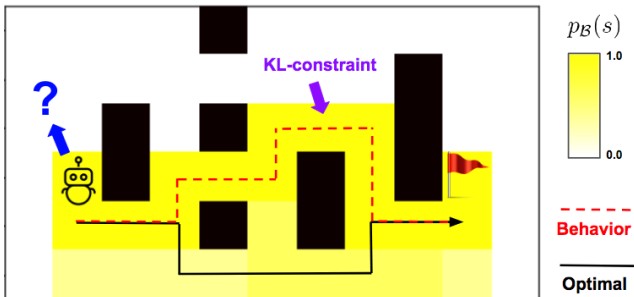

Figure 1: In this example batch RL problem, the robot's goal is to travel the minimum distance around the black walls to get to the red flag. A trained behavior policy generated the batch data; the probability of each of the states appearing in the batch, $p_\mathcal{B}(s)$, is in yellow (white locations are not contained in the batch). If the offline RL policy estimates the value of going *up* or *left* from the start position is high, it will have no way to refine this estimate using the batch data, or learn a good policy in this region of state space. The KL-constraint ensures that the RL policy will stay within the support of the batch data. However, the behavior policy is suboptimal, so using behavior cloning to directly imitate the batch data will result in suboptimal return. Instead, the KL-constrained model can learn to find the optimal policy, which is within the support of the batch.

extrapolation error. We also propose using dropout to obtain uncertainty estimates of the target $Q$-values, and use this lower bound to alleviate overestimation bias. We benchmark against a discrete adaptation of Batch Constrained $Q$-learning (BCQ) (Fujimoto et al., 2018b), a recently proposed state-of-the-art BRL algorithm for continuous domains, and show that our Way Off-Policy algorithm achieves superior performance in both a traditional RL domain, as well as in a challenging, under-explored, real-world reinforcement learning problem: using implicitly expressed human reactions in chat to improve open-domain dialog systems.

When a machine learning system interacts with humans, ideally we would like to learn about the humans' preferences in order to improve its performance. Yet having humans manually indicate their preferences through explicit means like pressing a button (e.g. Christiano et al. (2017)) or submitting a feedback report, does not scale. Instead, we would like to be able to use humans' implicit reactions, such as the sentiment they express, or the length of the conversation, in order to improve the policy. However, applying off-policy batch RL to language generation is challenging because the number of potential combinations of words and sentences leads to a combinatorial explosion in the size of the state space. The action space – the set of frequent vocabulary words in the English language – is 20,000-dimensional. This compounds extrapolation error, making BRL even more difficult. However, when learning from human interactions in the wild, it is crucial to be able to learn offline and test the policy before deploying it, lest it learn inappropriate behaviors (e.g. Horton (2016)).

To support this work, we developed an interactive online platform that allows humans to chat with deep neural network dialog models running on a GPU; the BRL models trained for this study are available live at `https://neural.chat/rl/`. Through this platform we collected human responses to a set of over 40 different dialog models over the course of several months. Using our Way Off-Policy algorithm, we are able to effectively learn from this batch of data, in spite of the fact that it was generated with a vastly different set of model architectures, which were trained on different datasets. Further, we use the batch to learn from many different reward functions designed post-hoc to extract implicit human preferences, something that is only possible with effective off-policy BRL.

In summary, the contributions of this paper are:

- A novel algorithm, Way Off-Policy learning, which is the first to propose using KL-control from a pre-trained prior model as a way to reduce extrapolation error in batch RL.
- Experiments showing the effectiveness of WOP above strong baselines based on prior work (e.g. Fujimoto et al. (2018b)), on both traditional RL tasks and on the challenging problem of open-domain dialog generation.
- A set of novel conversation rewards based on how human preferences are implicitly expressed in text. We are the first work to learn from implicit signals in conversation offline using batch RL.

## 2 RELATED WORK

The approach we propose is based on KL-control, a branch of stochastic optimal control (SOC) (Stengel, 1986) where the Kullback-Leibler (KL) divergence from some distribution is used to regularize an RL policy (e.g. (Abdolmaleki et al., 2018; Kappen et al., 2012; Rawlik et al., 2012; Todorov, 2007)). Well-known examples include Trust Region Policy Optimization (TRPO) (Schulman et al., 2015), and use conservative, KL-regularized policy updates to restrict the RL algorithm to stay close to its own prior policy (e.g. (Haarnoja et al., 2018; Kakade, 2002; Peters et al., 2010; Rawlik et al., 2012)). KL-control can also be applied to entropy maximization (e.g. (Ziebart et al., 2008; Nachum et al., 2017; Haarnoja et al., 2017)); for example, $G$-learning penalizes KL-divergence from a simple uniform distribution in order to cope with overestimation of $Q$-values (Fox et al., 2016).KL-control has also been used to improve transfer learning between maximum likelihood estimation (MLE) training on data, and training with RL (Jaques et al., 2017). To the best of our knowledge, our work is the first to propose penalizing KL-divergence from a learned prior model of the state-action space as a way to improve offline batch RL.

Other strategies to improve off-policy learning have been proposed, but differ from this work in key respects. Many focus on scenarios where the policy is able to explore and collect more data (e.g. Degris et al. (2012); Riedmiller (2005)); for example, when learning online from an outdated replay buffer (e.g. Munos et al. (2016)). In contrast, we learn entirely offline, from a fixed batch of data, without the ability to explore. Methods proposed for this setting have often not been used in conjunction with modern function approximation techniques (e.g. Thomas et al. (2015)). Many other works focus on off-policy policy *evaluation* (rather than policy learning), for example using importance sampling or model estimation (e.g. Farajtabar et al. (2018); Jiang & Li (2016); Precup (2000); Thomas & Brunskill (2016)). In the deep BRL setting, Liu et al. (2019) have proposed a correction to policy gradients, Gelada & Bellemare (2019) have proposed covariance-shift methods, and Bhatt et al. (2019) have proposed normalized feature representations.Kumar et al. (2019) use maximum mean discrepancy to cope with extrapolation error in BRL, while Agarwal et al. (2019) use a Random Ensemble Mixture (REM) $Q$-network. Most similar to our work is Batch Constrained Q-learning (BCQ) (Fujimoto et al., 2018b), which tackles off-policy deep BRL in continuous action domains by training a generative model of the batch, $p(a|s)$, sampling from this model, and selecting the best action based on a $Q$-estimate. Unlike our approach, this does not integrate information about the distribution $p(a|s)$ directly into the policy, or allow the model to learn when to strategically deviate from the prior in order to obtain more reward.

We propose using dropout to approximate model uncertainty of the target $Q$-network. The idea of using dropout to estimate uncertainty in neural networks was proposed by Gal & Ghahramani (2016). Different forms of uncertainty estimates have been used in RL (e.g. Kahn et al. (2017); Osband et al. (2016)); for example, Bayesian uncertainty estimates have been proposed as an alternative to double DQN (Azizzadenesheli et al., 2018).

### 2.1 RL FOR LANGUAGE GENERATION

Improving dialog systems with RL has largely been restricted to task-oriented dialog systems, which have a limited number of task-specific actions (e.g. Fatemi et al. (2016); Gašić et al. (2011); Liu & Lane (2017); Liu et al. (2018); Su et al. (2017)). These approaches may incorporate human input, usually through explicit, manual feedback (e.g. Shah et al. (2018)), but sometimes with more implicit signals, such as the user interrupting the system or starting over (Shi & Yu, 2018). Efforts to expand RL to the open-domain dialog setting, such as those of Li et al. (2016b; 2017; 2018), are less numerous, and do not involve learning from human feedback. Even in the open-domain setting, authors may choose to use a highly restricted action space; for example, using RL to choose which scripted or MLE dialog model to invoke to answer a user's query (Serban et al., 2017a).

Since the posting of the preprint of this paper, Ziegler et al. (2019) have used explicit human feedback to improve the summarization and text continuation performance of a large-scale language model. Although they do not study dialog or the batch RL setting (instead learning online from a trained model of human feedback), they do make use of our proposal to penalize KL-divergence from a pre-trained language model, and find that this is important to achieving good performance.

Although implicit signals such as sentiment (Hancock et al., 2019) and conversation length (Zhou et al., 2018) have been used in MLE systems, the idea of using such signals as a reward for RL is relatively unexplored. Shin and colleagues uses on-policy learning in conjunction with a user-sentiment approximator to improve a seq2seq model (Shin et al., 2019), but are unable to learn directly from user feedback. To the best of our knowledge, we are the first to use batch RL to train open-domain dialog models on implicit cues gained from real human interactions.

## 3 METHODS

We employ typical RL notation in which $s_t$ represents the environment state at time $t$, the agent takes action $a_t$ according to its policy $\pi(a_t|s_t)$, and receives reward $r(s_t, a_t)$. The agent's goal is to maximize reward over an episode trajectory $\tau$, with a discount factor of $\gamma$ applied to future rewards. $Q$-learning learns an action-value estimate of the total expected discounted future reward, $Q_\pi(a_t, s_t) = \mathbb{E}_\pi[\sum_{t'=t}^{T} \gamma^{t'-t} r(s_{t'}, a_{t'})]$, through iterative updates based on the Bellman equation:

$$Q_{\theta_\pi}(s_t, a_t) = r(s_t, a_t) + \gamma \mathbb{E}_{s_{t+1} \sim p(\cdot|s_t, a_t)}[\max_{a_{t+1}} Q_{\theta_T}(s_{t+1}, a_{t+1})] \tag{1}$$

In deep $Q$-learning (Mnih et al., 2013), a $Q$-network approximates $Q_{\theta_\pi}(s_t, a_t)$ and drives the policy $\pi$. A second target $Q$-network approximates the expected reward from the next state, $Q_{\theta_T}(s_{t+1}, a_{t+1})$ (Van Hasselt et al., 2016).

### 3.1 BATCH RL AND EXTRAPOLATION ERROR

In batch RL, we are given a fixed batch of data $\mathcal{B}$, and assume that no further interaction with the environment is possible. To train $Q_{\theta_\pi}$, we sample $(s_t, a_t, r_t, s_{t+1}) \sim \mathcal{B}$, and update the weights of the $Q$-network to approximate Eq. 1. Because $Q$-learning is an off-policy algorithm, in principle it should be able to learn from data collected by any behavior policy. However, extrapolation error can occur if the BRL policy learns to favour a state-action pair $(s, a)$ that is unlikely, or not contained, in the batch data. In this case, the estimate $Q(s', \pi(s'))$ can be arbitrarily bad (Fujimoto et al., 2018b). Such errors can then accumulate through the Bellman backup operator (Kumar et al., 2019). Experiments from Fujimoto et al. (2018b) show that extrapolation error can be highly detrimental to learning off-policy in BRL.

These problems are compounded by the fact that algorithms based on the Bellman operator are inherently optimistic in the face of uncertainty. When value estimates for some region of the state-action space are noisy (because too few experience samples have been used to refine them), the maximum operation in Eq. 1 will lead to an overestimation of expected future reward. In a normal RL setting, this overestimation bias drives the model to explore areas of the state-action space for which the value estimates have the highest variance, thus enabling it to refine them; in essence, creating a built-in drive to explore. However, in a batch setting where exploration is not possible, the model is instead driven to value parts of the state-action space for which it has little to no data to learn a good policy (see Figure 1).

### 3.2 DROPOUT FOR UNCERTAINTY ESTIMATION OF TARGET $Q$-VALUES

The overestimation of $Q$-values in the BRL setting necessitates other methods for estimating the future reward via the Target $Q$-network. Clipped Double $Q$-learning (Fujimoto et al., 2018a) maintains two independent pairs of $Q$-networks, and takes the minimum of their estimates of future reward. This approach is computationally expensive and memory intensive. Instead, we leverage the fact that a network trained with dropout can be used to approximate a Bayesian uncertainty estimate of the output value (Gal & Ghahramani, 2016). Given the target $Q$-network $Q_{\theta_T}$, we compute $Q(a_{t+1}, s_{t+1})$ using a Monte Carlo (MC) estimate of the lower-bound of $Q_{\theta_T}(a_{t+1}, s_{t+1})$ by running $M$ stochastic forward passes of the network, each with a new dropout mask $d_i \sim q^W$:

$$Q(a_{t+1}, s_{t+1}) = \min_{i=1...M}[Q_{\theta_T}(a_{t+1}, s_{t+1}; d_i)] \tag{2}$$

Using the minimum operator penalizes high variance estimates and leads the algorithm to be pessimistic in the face of uncertainty, rather than optimistic. Such a bias will push the model to favour actions that lead to states well covered by the batch data.

### 3.3 DISCRETE BATCH CONSTRAINED $Q$

Batch Constrained Q-learning (BCQ) (Fujimoto et al., 2018b) proposes to address the BRL problem by constraining the actions of the $Q$-network to be close to the data contained within the batch. This is accomplished by learning a generative model of the batch, $G_w = p(a|s)$, and sampling from this model during learning and inference. Because BCQ is designed for continuous action domains, it applies a learned perturbation model $\xi(s, a; \Phi)$ which is allowed to alter the action within the range $[-\Phi, \Phi]$. BCQ learns $Q$-estimates that incorporate the perturbation model, $Q_\theta(s, a + \xi(s, a; \Phi))$. To act, $n$ possible actions are sampled from the generative model, $\{a_i \sim G_w(s)\}_{i=1}^n$, perturbed, and the action with the maximum $Q$-value is selected, giving the BCQ policy:

$$\pi_{BCQ}(s) = \underset{a_i + \xi(s, a_i; \Phi)}{\arg\max} Q_\theta(s, a_i + \xi(s, a_i; \Phi)) \tag{3}$$

We propose an adaptation of BCQ to discrete action spaces (*DBCQ*) which does not use a continuous perturbation model. Since BCQ relies on Double Clipped $Q$-learning (Fujimoto et al., 2018a), here we use dropout-based uncertainty estimates as in Eq. 2. Thus the DBCQ policy is:

$$\pi_{DBCQ}(s) = \underset{a_i \sim p(a|s)}{\arg\max} Q_{\theta_\pi}(s, a_i) \tag{4}$$

### 3.4 KL-CONTROL FROM PRE-TRAINED PRIOR

Rather than simply sample from the prior, we would like the $Q$-learning algorithm to directly incorporate the prior into the policy. Thus, we use KL-control to penalize divergence between the learned prior $p(a|s)$, and the $Q$-network policy $\pi_\theta$, while still maximizing reward. Given a trajectory of actions, $\tau = \{a_1, a_2, ...a_{t-1}\}$, let $q(\tau) = \prod_{t=1}^T \pi_\theta(a_t, s_t)$ be the policy of our $Q$-learning algorithm at the trajectory level. Similarly, let $p(\tau) = \prod_{t=1}^T p(a_t|s_t)$ be the prior distribution over the trajectory, and $r(\tau)$ be the return. We seek to maximize the following KL-regularized objective:

$$L(q) = \mathbb{E}_{q(\tau)}[r(\tau)]/c - D_{KL}[q(\tau)||p(\tau)] \tag{5}$$

Since $D_{KL}[q||p] = \sum_x q(x)(\log q(x) - \log p(x))$, we can see that this is equivalent to maximizing the following expected value function of the policy $\pi_\theta$ at the action level:

$$Q^\pi(s_t, a_t) = \mathbb{E}_\pi[\sum_{t'=t}^T r(s_{t'}, a_{t'})/c + \log p(a_{t'}|s_{t'}) - \log \pi(a_{t'}|s_{t'})] \tag{6}$$

The two terms introduced in Eq. 6 have clear motivations. The $p(a|s)$ term rewards the model for choosing actions that have high probability under the prior, biasing the model to state-action pairs that are likely to be in the batch. The $-\log \pi(a|s)$ term is analogous to entropy regularization. Maintaining diversity in the action space through entropy regularization is important for generative models like dialog systems, which are known to collapse to an uninteresting, small number of repeated samples (Li et al., 2016a). Re-stating Eq. 6 as an entropy-regularized $Q$-function, we obtain:

$$Q(s_t, a_t) = \mathbb{E}_\pi[\sum_{t'=t}^T r(s_{t'}, a_{t'})/c + \log p(a_{t'}|s_{t'}) + \mathcal{H}(\cdot|s_{t'})] \tag{7}$$

One can derive a soft version of the entropy-regularized $Q$-function that uses a Boltzmann distribution to estimate future reward (Haarnoja et al., 2017). We refer to it as a $\Psi$-function following previous work (Jaques et al., 2017), which derived this function as a generalization of the $\Psi$-learning proposed by (Rawlik et al., 2012). The optimal $\Psi$-function and policy are:

$$\Psi^*(s_t, a_t) = r(s_{t'}, a_{t'})/c + \log p(a_{t'}|s_{t'}) + \gamma \log \sum_{a'} \exp(\Psi^*(s', a')) \tag{8}$$

$$\pi_\Psi^*(a_t|s_t) = \exp(\Psi^*(s_t, a_t)) \tag{9}$$

Because it avoids taking a hard max over noisy estimates, $\Psi$-learning leads to less overestimation of future reward (Abdolmaleki et al., 2018; Haarnoja et al., 2017). This improves learning through more stable temporal-difference (TD) updates. Thus, it may be especially useful in the BRL setting for reducing optimism in the face of uncertainty. The Way Off-Policy (WOP) algorithm combines Monte Carlo (MC) target estimation, $\Psi$-learning, and KL-control from a pre-trained prior.

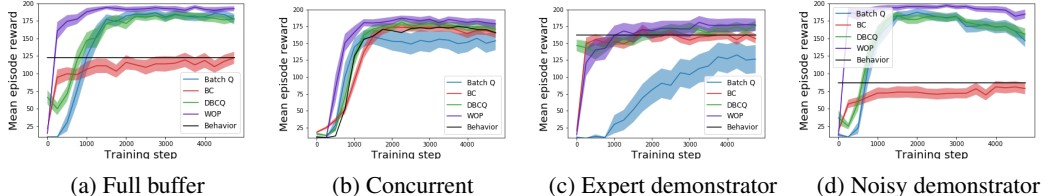

|(a) Full buffer|(b) Concurrent|(c) Expert demonstrator|(d) Noisy demonstrator|

Figure 2: Comparison of batch RL algorithms for different offline learning conditions. WOP consistently exceeds the performance of Batch Q-learning, Behavioral Cloning (BC), DBCQ, and the Behavior policy used to generate the batch data. Error bars show 95% *CI* of the mean over 50 trials.

## 4 TRADITIONAL RL EXPERIMENTS

To demonstrate the effectiveness of these techniques, we conduct a series of experiments in traditional RL tasks using the OpenAI gym (Brockman et al., 2016). Here we show results for the *CartPole-v0* environment; more results are available in the Appendix. We first train an online $Q$-learning *Behavior* policy, and store all $(s, a, r, s')$ experience samples into a replay buffer. We use this buffer to train a prior model of $p(a|s)$ using a Variational Auto-encoder (VAE) (details in Appendix). This model is used as a part of both the DBCQ and WOP algorithms. We can use the prior for imitation learning, by sampling actions directly from $p(a|s)$ to obtain Behavioral Cloning (BC). We benchmark all of these techniques against vanilla $Q$-learning on the batch data (*Batch Q*).

We experiment with four different conditions which vary the quality of the Behavior policy and the replay buffer data: a) *Full buffer*: all experience samples experienced during online training are used for offline learning; b) *Concurrent*: the offline learning algorithms see a sliding window of experience samples in the same order that the online learner experienced them; c) *Expert demonstrator*: the buffer only contains experience generated by a fully trained online learner; and d) *Noisy demonstrator*: the online learner has a high probability of acting randomly ($\epsilon = 0.3$) and is thus a bad model of the optimal policy.

Figure 2 shows the results. Across conditions, we see that WOP is able to outperform Batch $Q$, imitation learning (BC), DBCQ, and the original behavior policy. As expected, Imitation learning (BC) underperforms other techniques when the batch contains noisy or inexpert experience samples. However, when the batch contains only expert trajectories, Batch $Q$ fails to learn, because the batch does not cover the full state-action space well, increasing extrapolation error (as illustrated in Figure 1). DBCQ matches or outperforms BC and Batch $Q$ in all scenarios. However, because DBCQ acts by sampling from $p(a|s)$ as learned by the BC model, its performance suffers when the batch data is noisy or imperfect. In contrast, WOP is able to learn to trade-off staying close to the prior and obtaining higher reward, and consistently outperforms all other algorithms in this environment.

## 5 BATCH RL FOR LEARNING DIALOG FROM HUMAN FEEDBACK

Here, we tackle the problem of training an open-domain dialog model from human feedback. We consider human interaction to represent the 'environment'. The response of a human to the bot's utterance is used to compute a reward signal to train the model. The state is the conversation history, composed of a series of conversation turns or utterances, $u_{1...t}$, where each utterance is composed of vocabulary tokens. The model attempts to construct a response utterance $u_{t+1}^{\pi} = [a_1, a_2, ..., a_n]$ by iteratively choosing an action $a_i$ as the next token. Applying RL to dialog generation is challenging due to the large state-action space. The number of tokens in the vocabulary of our pre-trained model is 20,000, making the action space very high-dimensional; this further compounds the problem of extrapolation error.

We trained over 40 dialog models with different architectures (e.g. Serban et al. (2017b)), on different datasets, generating models that varied significantly in terms of the distribution of language they learned. We deployed these models to users via a web server that hosts neural network dialog models on GPU for fast, real-time inference: `https://neural.chat`. The code for the models and the server is available in open-source at *<redacted>*. Using the server, we collected a batch

of human interaction data containing 14232 pairs of user input and agent response. Because learning language online from humans on the internet can result in inappropriate behavior (see Horton (2016)), learning offline using BRL is imperative.

The batch data was used to train the RL models as described in Section 3. Here, we use a pre-trained language model to estimate $p(a|s)$. We also initialize the weights of the $Q$-network and target $Q$-network are from the pre-trained model, to combat extrapolation error. The trained RL models were then re-deployed to the web. We recruited 90 Mechanical Turk workers to provide a total of 718 7-point Likert scale ratings of the bots' quality, fluency, diversity, contingency (relatedness), and empathy, after interacting with each bot for at least 3 turns. Participants also had the option to provide explicit feedback through upvoting or downvoting a particular utterance within the interface. We sum these manual votes to create an overall *votes* score. We note that using this platform to test our models "in the wild" with humans represents a more meaningful test of generalization than testing an RL model in the same limited (game) environment in which it was trained, since humans are not restricted in the text they can type as input to the model.

### 5.1 LEARNING FROM IMPLICIT HUMAN PREFERENCES

We seek to improve a dialog model's ability to engage in natural conversation with a human by learning from the signals implicit in the way that the human responds. Rather than having the human manually label good performance – which we show in this work does not scale – the agent should recognize informative cues within the user's responses, like sentiment, and the amount of time they spend chatting. Essentially, we want to create an agent that is intrinsically motivated to produce positive reactions in its human conversation partner. To this end, we reward the model for: 1) eliciting positive sentiment, 2) eliciting longer conversations and more words typed (a sign of engagement), 3) eliciting laughter (in the form of typed 'ha's), 4) high semantic similarity between the human input and bot response, and 5) asking questions, since this is an important active listening skill (Bodie et al., 2012). The total reward given to the agent is a combination of these, with details (and coefficients) in the Appendix. Note that the first 4 types of rewards depend on eliciting positive responses from a human user; we call these the implicit *human reward*. The 5th reward is easily exploitable by the agent itself. These rewards were designed and extracted post-hoc from the batch of human data, and thus learning from them is only possible with effective batch RL, since they had no effect on the policies used to generate the batch.

## 6 DIALOG RESULTS

Table 1: Interactive human evaluation of batch RL techniques. KL-control models strongly outperform other techniques. Ratings are Likert scale, votes and human reward are $z$-scores.

| Model type | Quality | Fluent | Diverse | Related | Empathy | Total | Votes | Human reward |
|---|---|---|---|---|---|---|---|---|
| DBCQ | 1.64 ±.29 | 1.87 ±.34 | **3.13 ±.58** | 1.84 ±.34 | 2.09 ±.38 | 10.58 ±1.55 | -.228 | -.050 |
| Batch Q | 1.87 ±.30 | 2.36 ±.42 | 2.20 ±.41 | 1.91 ±.32 | 2.58 ±.47 | 11.91 ±1.58 | -.163 | -.005 |
| Batch Q + MC | 1.85 ±.39 | 2.46 ±.44 | 2.46 ±.52 | 1.98 ±.39 | 2.34 ±.49 | 11.07 ±1.82 | -.068 | .005 |
| KL-control Q | **2.38 ±.39** | 3.24 ±.47 | 3.42 ±.54 | **2.38 ±.45** | 2.56 ±.43 | 13.98 ±1.81 | .016 | .004 |
| KL-control $\Psi$ (WOP) | 2.33 ±.41 | **3.73 ±.53** | 2.82 ±.50 | 2.31 ±.44 | **3.47 ±.50** | **14.67 ±1.82** | **.128** | **.061** |

Table 2: Purely reward-maximizing methods like Batch $Q$ (left) diverge away from realistic language (saying phrases like *"where did you say to me?"*) in order to trivially exploit the reward function by asking a question every turn, and using the maximum number of tokens in every sentence. In contrast, KL-control methods (right) output plausible language by staying close to the prior, but shift to using polite, cheerful language to maximize implicit human reward.

**[User]**: hey, what's up?
**[Batch Q]**: where did you get??, that sounds so exciting. how do you work for you... you want to talk to me?... you want
**[User]**: yes, i want to talk to you!
**[Batch Q]**: where did you say to me?... you want to talk to you about you... you are so good for you... you are so good...
**[User]**: thanks, i appreciate that.
**[Batch Q]**: where did you get??, that sounds so exciting. how do you work for you... you want to see you...

**[User]**: hey, what's up?
**[KL-control]**: i'm not sure, but i'm glad to see you, and i'm so happy that you're not alone
**[User]**: oh thanks, that's very nice of you!
**[KL-control]**: i'm sorry, but i'm glad to see what you think.
**[User]**: don't be sorry, you did a good job.
**[KL-control]**: that's really nice to me.

To compare models, we not only look at human users' ratings and votes, but also consider the automatic signals detectable from the text itself. This implicit *human reward* metric aggregates the measures listed in items 1-4 in Section 5.1, and measures the ability to elicit positive responses from a human. Table 1 shows the results of the human evaluation, comparing WOP to ablations of itself, Batch Q, and DBCQ. MC Target $Q$ estimation leads to modest improvements in *votes* and *human reward*, but does not improve ratings. Using $\Psi$-learning improves all three. However, the most notable difference in performance comes from KL-control. The KL-control models show substantial gains over the baseline models across both ratings and human reward. We perform a one-way analysis of variance (ANOVA) comparing the KL-control models to the Batch $Q$ baselines and DBCQ on the total human rating score, and find that the KL-control models are significantly better, $F(x) = 4.781, p < .05$. This validates the hypothesis that KL-control with a strong, pre-trained prior can be used to improve batch RL.

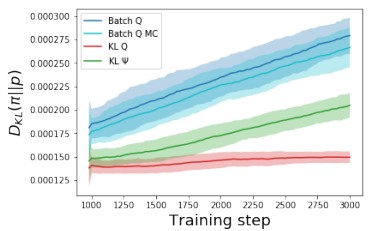

Figure 3: KL-divergence of the policy from the prior is lower with KL-control throughout training. Bands show $\sigma$.

As shown in Figure 3, without KL-regularization, the baseline RL models diverge quickly and continuously from the prior, losing information about realistic sequences. This helps explain the poor performance of DBCQ in Table 1. The underlying $Q$-network in DBCQ does not directly integrate the prior. As $Q$-learning causes the model to diverge from the prior, the $Q$-estimates of language generated according to the prior become unrealistic, and Eq. 4 selects unrealistic actions. This results in highly 'diverse' (random) generated utterances. Although DBCQ performed well in simple domains in Section 4, it does not scale effectively to dialog.

The pre-trained prior may be especially important in a generative domain like dialog, where the true reward function is unknown, and so purely maximizing a heuristic reward may lead to lower quality conversations. Table 2 shows examples of conversations with a Batch $Q$ and KL-control model. Because the Batch $Q$ model has no incentive to stay close to realistic language, it learns to exploit the reward by asking a question and outputting the maximum number of tokens (30) every utterance. These sentences contain implausible phrases that do not represent realistic language (e.g. "*where did you say to me?*"). In contrast, the KL-control model uses fluent language, but shifts its distribution towards cheerful and polite speech, presumably because this is what led to positive human responses in the batch data.

In fact, we noticed that all models trained with the implicit human rewards described in Section 5.1 learned to use more cheerful and supportive language. Therefore, we create post-hoc metrics to measure this effect (see the Appendix for details). Figure 4 shows how these metrics, as well as the implicit rewards, differ across models. Without KL-control, baseline methods like Batch Q exploit simple rewards like asking questions at the expense of realistic language, explaining their poor quality ratings. In contrast, KL-control models learn to rely more on realistic but polite, supportive, and cheerful dialog to elicit higher total *human reward*.

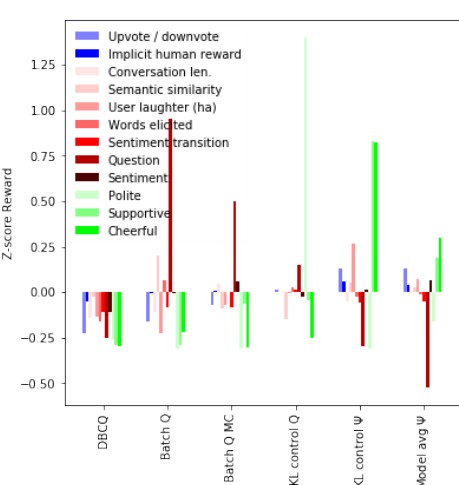

Figure 4: Z-scored reward. Red metrics were used in training rewards, green are post-hoc. Traditional RL methods like Batch Q exploit simple action-based rewards, like asking questions. In contrast, KL-control methods shift their distribution towards polite, supportive, and cheerful conversation, allowing them to elicit higher *human reward* (blue).

Table 3 presents the results of WOP models trained with only a single reward function, ordered from lowest to highest quality. Notably, extracting multiple different reward functions post-hoc from a batch of data and training on these independently is only possible with effective BRL. Investigating which rewards presented are most critical to achieving high-quality conversations with humans, we note that maximizing positive and minimizing negative sentiment in the user turns out to lead

Table 3: Interactive human evaluation of different reward functions (models trained with WOP).

| Reward function | Quality | Fluent | Diverse | Related | Empathy | Total | Votes | Human reward |
|---|---|---|---|---|---|---|---|---|
| Conv. len. | 2.20 ±.40 | 3.61 ±.53 | 3.02 ±.52 | 2.25 ±.46 | 2.48 ±.45 | 13.57 ±1.84 | -.035 | -.003 |
| Semantic sim. | 1.93 ±.34 | 3.50 ±.45 | 2.37 ±.45 | 2.11 ±.45 | 2.52 ±.48 | 12.43 ±1.75 | -.020 | .012 |
| User laughter | 1.96 ±.38 | 3.56 ±.48 | 2.33 ±.51 | 1.93 ±.42 | 3.20 ±.55 | 12.98 ±1.60 | -.149 | -.003 |
| Words elicited | 2.11 ±.32 | 3.96 ±.44 | 3.04 ±.45 | 2.04 ±.35 | 2.55 ±.46 | 13.70 ±1.44 | .059 | .024 |
| Manual votes | 2.14 ±.38 | 3.47 ±.45 | 2.91 ±.47 | 2.07 ±.39 | 2.42 ±.46 | 13.00 ±1.65 | -.030 | .010 |
| Sent. trans. | 2.02 ±.31 | 3.71 ±.49 | 2.98 ±.50 | 2.04 ±.42 | 2.84 ±.48 | 13.60 ±1.63 | .031 | .014 |
| Question | 2.29 ±.37 | **4.31 ±.50** | **3.31 ±.52** | 2.20 ±.40 | 2.60 ±.41 | 14.71 ±1.63 | .057 | .012 |
| Sentiment | **2.47 ±.32** | 4.05 ±.45 | 3.23 ±.46 | **2.42 ±.39** | **3.23 ±.55** | **15.40 ±1.49** | **.085** | **.045** |

to the highest quality bot. This underscores the importance of affective signals as cues for good conversation. Bots trained on the manual upvotes and downvotes provided by users on the utterance level fail to achieve similarly high performance. Even though users were instructed to make use of the vote feature, the task is burdensome, and users did not vote frequently enough to provide a good training signal. This validates the hypothesis that *implicit* signals of human enjoyment (such as sentiment) are a more scalable way to learn from human preferences.

## 7 CONCLUSION

This paper presents the Way Off-Policy (WOP) algorithm, which improves performance when learning off-policy without the possibility to explore – i.e. batch RL (BRL). We are the first to propose using KL-control from a strong prior model pre-trained on data as a way to avoid extrapolation and instability in BRL. Our results on traditional RL tasks demonstrate that our WOP algorithm provides performance improvements over state-of-the-art BRL techniques, and the results in dialog generation show that KL-control is critical to achieving good performance in this real-world, high-dimensional setting. In a generative domain such as dialog, the true reward function is not known, and trivially exploiting the rewards can actually lead to worse performance. Thus, KL-control may be particularly necessary to ensure samples remain realistic and close to the data distribution. We propose several reward functions that could allow an open-domain dialog generation model to learn from rich cues implicit in human interaction, where learning from expressed sentiment was most promising. We find that maximizing implicit rewards leads to better performance than relying on explicit feedback. We hope that the techniques presented here can improve learning with RL from offline data, making it easier to apply RL to safety-critical settings such as human interaction.

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

# A    APPENDIX

## A.1    DETAILS ABOUT IMPLICIT METRICS

The total reward used to train the bots is a combination of the rewards described below, in the following proportions:

```
0.15682657*question + 0.13837638*semantic_coherence +
0.15313653*laughter + 0.14206642*sentiment_transition
+ 0.14206642*sentiment + 0.14760148*words_elicited +
0.1199262*conversation_length.
```

### A.1.1    SENTIMENT-BASED

To compute sentiment on short texts like conversation utterances, we leverage a state-of-the-art sentiment-detection model, which was trained on a massive amount of Twitter data to predict the emojis in tweets (Felbo et al., 2017). Transfer learning from this model to other tasks showed that it was able to significantly outperform a series of sentiment, irony, and sarcasm benchmarks. This DeepMoji model outputs a probability distribution over 64 most-frequently used emojis as shown in Figure 5. After observing the performance of the model in detecting users' emotions in the domain of online chat, we define a set of weights over the emojis and calculate the weighted sum over an emotion embedding vector to derive a *sentiment* reward which is higher for positive sentiment and lower for negative sentiment. These weights are shown in Figure 5 (b). We also compute a sentiment-transition reward using the same score based on whether the peak positive sentiment occurred later in the conversation than the peak negative sentiment, reasoning that sentiment should improve over the course of the conversation.

### A.1.2    ENGAGEMENT-BASED

Based on prior work (Zhou et al., 2018), we use the number of turns in the conversation as an indicator of the quality of the bot's performance. To distribute this reward over every utterance in the conversation, we take the total conversation length $N$, and compute the discounted reward for utterance $n < N$ as $\gamma^{N-n}N$. We also reward each utterance with the number of words in the user's response, which we refer to as the *words elicited*.

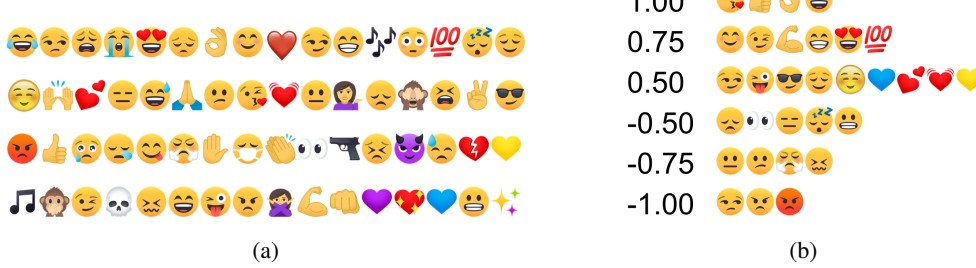

Figure 5: (a) 64-most frequent emojis as predicted by (Felbo et al., 2017) used for calculating emotion embeddings. (b) Assigned weights used in producing the sentiment reward from the predicted emoji values.

### A.1.3 LAUGHTER

Laughter has been shown to be very important to human affiliation (Provine, 1996) and solidarity (Hay, 2000). Therefore, we detect the number of occurrences of the string 'ha' in the user's response, and use this as a reward. Interestingly, we find that bots trained to maximize user laughter learn to be extremely supportive and cheerful compared to other bots (for definitions of supportive and cheerful, see Section **??**).

### A.1.4 SEMANTIC SIMILARITY

Language style matching has been shown to be a strong predictor of relationship initiation and stability (Ireland et al., 2011). While it would be ideal if our chatbots could intelligently adapt their conversation style to a new user, in reality most baseline dialog models struggle to maintain topic coherence, even over a few utterances (for an analysis of this effect, see (Ghandeharioun et al., 2019)). Therefore we reward *semantic similarity* between the user's input and the bot's response, to encourage the bot to stay on topic and produce reasonable answers. This score is computing by leveraging a state-of-the-art sentence embedding model (Conneau et al., 2017), and penalizing distance in embedding space.

### A.1.5 QUESTIONS

Asking questions is an important listening skill, and is linked to conversation management, attentiveness, and responsiveness (Bodie et al., 2012). Therefore, we give the bot a reward of 0.5 if the utterance contains a question word (*how, what, where, why, when, who*), and an additional 0.5 if it contains a question mark.

### A.1.6 POST-HOC METRICS

After training the bots on these rewards, we noticed a shift in the distribution of their language towards more polite, cheerful, and supportive speech. Therefore, we designed post-hoc metrics to measure these qualities, which are based on counting whether a subset of phrases is present in an utterance.

**Politeness phrases:** *if I may; may I; please; thanks; no worries; if you don't mind; have a great day; I'm sorry*.

**Supportive phrases:** *you're right; you are right; you're not alone; you are not alone; congrats; that's a good idea; that is a good idea; you'll be fine; you will be fine; you'll be okay; you will be okay; it will get better; sorry you're going through; sorry you are going through; if it makes you feel better; if it makes you feel any better; keep your head up; keep it up; I'm in a similar situation; I am in a similar situation; you'll get it; you will get it; happy for you; I'm in the same boat; I am in the same boat; if you feel like you need to vent*.

**Cheerful phrases:** *nice to hear; happy; excited; really nice; glad; the best; great; good time; looking forward; beautiful*.

### A.2 Training details and hyperparameters

#### A.2.1 Traditional RL experiments

All $Q$-networks shared the same underlying architecture: three fully-connected layers of size [256, 128, 64], with ReLU activation between. The model of $p(a|s)$ was learned with a Variational Auto-encoder which attempted to reconstruct the next state given the current state, $p(s'|s)$, using a mean-squared error loss. Both the encoder and decoder were made up of two linear layers with 750 neurons each. The latent dimension of the VAE was size 256. The next action, was predicted from the latent embedding $z$, meaning the model learned three functions: $z = f_e(s)$, $s' = f_d(z)$, and $a = f_a(z)$ All models were trained with the Adam optimizer Kingma & Ba (2014).

For each experiment, we ran 50 trials of each model with a different random seed each time. The Behavior policy was trained for a total of 20,000 steps in the environment, so in the *Full buffer* condition offline agents saw 20,000 experience samples. The Behavior policy typically converged before 10,000 steps, so in the *Expert demonstrator* condition the offline agents received the last 10,000 experience samples from the trained agent. In the *Concurrent* condition, offline agents saw a moving window of 1000 samples, since the online learner only used the most recent 1000 samples in the buffer for learning. The learning rate was .001, $\gamma = .99$, and $\epsilon$ decayed linearly from 1.0 to .01 over 2000 steps. The KL-constraint was computed as $D_{KL}[q(\tau)||p(\tau)] = \alpha \log p(a|s) - \beta \log \pi(a|s)$, where $\alpha = 0.5$ and $\beta = 0.1$. DBCQ sampled $n = 2$ actions before selecting the best action based on the maximum $Q$-value; note that in this environment there are only 2 actions.

#### A.2.2 Dialog experiments

The underlying architecture of the language models employed for this work is a Variational Hierarchical Recurrent Encoder Decoder (VHRED) (Serban et al., 2017b), with additional knowledge distillation to improve the model's ability to track the sentiment and semantics of the conversation, as proposed by Ghandeharioun et al. (2019). The language models were originally trained on two datasets: movie dialogs (Danescu-Niculescu-Mizil & Lee, 2011) and a dataset scraped from `reddit.com/r/casual_conversation` (Ghandeharioun et al., 2019). The RL models were initialized with the weights of the best model trained on the Reddit dataset.

RL models were trained for between 800 and 1000 batches of data, where the batch size was fixed at 32. Early stopping was used to determine the number of training iterations of the best checkpoint. All other hyperparameters were shared between RL models, and were as follows: discount $\gamma = 0.5$, weight placed on RL reward vs. KL-divergence term $c = 2$, number of Monte Carlo samples of the Target $Q$-network $M = 5$, target network update rate $\alpha = .005$, learning rate $r = .0001$. We used a smooth $L1$ loss function to approximate the $Q$-values, and clipped gradients at a value of 1.0.

The underlying parameters of the VHRED model were as follows: Context RNN hidden size $= 1000$, decoder hidden size $= 1250$, encoder hidden size $= 1250$, $z$ embedding size $= 600$, gradient clip $= 1.0$, dropout $d = 0.2$. The maximum conversation length was fixed at 5 utterances (context from more than 5 utterances ago was discarded), and the maximum sentence length was 30 tokens.

We also added layers to the Context RNN and regularized it to be able to predict the semantic content of the input utterance using a form of knowledge distillation (Hinton et al., 2015) from a state-of-the-art sentence-embedding model (Conneau et al., 2017). There were 2 additional feedforward semantic prediction prediction layers of size 128, which used ReLu activation.

### A.3 Additional results

#### A.3.1 Traditional RL experiments

We ran additional experiments in another standard OpenAI gym environment, *Acrobot-v1*; Figure 6 shows the results. The experimental setup and hyperparameters remained the same as in the first experiment, except that we used a smaller VAE (the encoder and decoder had only one layer of size 256 each, and the latent dimension was 64), the weight on the prior was $\alpha = 0.5$ and the entropy term was $\beta = 0.1$, and we used the $Q$-learning loss rather than $\Psi$-learning.

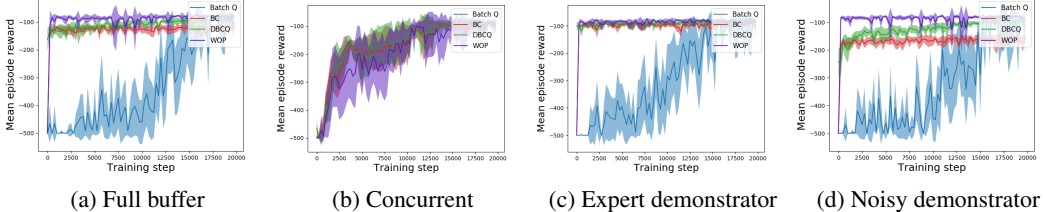

(a) Full buffer     (b) Concurrent     (c) Expert demonstrator     (d) Noisy demonstrator

Figure 6: Comparison of batch RL algorithms for different offline learning conditions in *Acrobot-v1*. WOP exceeds the performance of Batch Q-learning, Behavioral Cloning (BC), DBCQ, and the Behavior policy used to generate the batch data. Error bars show 95% *CI* of the mean over 10 trials.

| | Total reward | Implicit human reward | Conversation length | Sentiment similarity | User laughter (ha) | Words elicited | Sentiment transition | Question | Sentiment | Manual ratings | Politeness | Supportive | Cheerful |
|---|---|---|---|---|---|---|---|---|---|---|---|---|---|
| Conversation length | 0.018 | -0.003 | -0.046 | -0.048 | 0.023 | -0.053 | 0.008 | -0.541 | -0.102 | -0.035 | -0.037 | -0.143 | -0.277 |
| Semantic similarity | 0.037 | 0.012 | -0.003 | -0.145 | 0.119 | -0.039 | 0.006 | -0.480 | -0.042 | -0.020 | -0.191 | -0.224 | 0.068 |
| User laughter (ha) | 0.029 | -0.003 | -0.101 | 0.060 | 0.023 | -0.016 | -0.105 | -0.368 | -0.056 | -0.149 | -0.210 | 0.886 | 0.826 |
| Words elicited | 0.072 | 0.024 | 0.135 | 0.043 | -0.093 | 0.003 | -0.038 | -0.126 | 0.014 | 0.059 | 0.570 | -0.285 | -0.016 |
| Sentiment transition | 0.051 | 0.014 | -0.059 | 0.034 | 0.100 | -0.036 | -0.004 | -0.299 | -0.123 | 0.031 | 0.103 | 0.133 | 0.015 |
| Question | 0.154 | 0.012 | 0.195 | -0.012 | -0.230 | 0.047 | -0.048 | 1.366 | 0.083 | 0.057 | -0.210 | -0.364 | -0.251 |
| Sentiment | 0.123 | 0.045 | -0.092 | 0.021 | 0.030 | 0.068 | -0.099 | 0.346 | 0.294 | 0.085 | 0.098 | 0.270 | -0.073 |
| Manual ratings | 0.064 | 0.010 | -0.065 | 0.051 | 0.062 | 0.026 | -0.071 | -0.029 | -0.064 | -0.030 | -0.112 | -0.207 | -0.275 |

Figure 7: Normalized reward scores obtained by models trained with respect to different rewards. We see that the bot trained to ask questions is easily able to exploit this reward, and similarly the bot trained to elicit positive sentiment does so successfully. For the rest of the bots, the relationship is less clear. For example, the bot trained to elicit laughter becomes the most supportive and cheerful, while the bot trained to elicit more words is very polite.

### A.3.2 DIALOG EXPERIMENTS

Figure 7 shows the normalized reward scores obtained bots trained with respect to different rewards. We see that in testing the bots in the wild, with a new set of users, bots trained to optimize certain types of human response (e.g. words elicited) do not perform best on this task. We hypothesize this is because the relatively small size of batch date we were able to collect ($\approx 14,000$ utterances) does not give the bots enough information about how to elicit long responses from users.

In addition to the comparisons between RL models, we also benchmarked the RL methods against the MLE prior. While we found that the KL-control methods exceed the prior in obtaining high implicit reward signals from humans (e.g. eliciting positive sentiment, laughter, more words, etc), we found that this did not lead to them being rated significantly higher in quality by human judges. Even though the RL bots successfully learned to conduct conversation to optimize these implicit signals, we believe the fact that this did not lead to higher quality ratings highlights the idea that the reward functions we are optimizing do not fully cover what it means to have a high quality conversation with a human user. We note that these rewards are only a first step, and hope other researchers will be able to use the techniques proposed in this paper to learn from better measures of human enjoyment.

## B    INTERACTIVE BOT PLATFORM DETAILS

To collect data from humans interacting with our bots, we built `https://neural.chat`, a platform for hosting deep neural network dialog models online on GPU for fast, real-time inference. Figure 8) shows an example of the interface, in which users are able to rate the bots after talking to them for at least three turns.

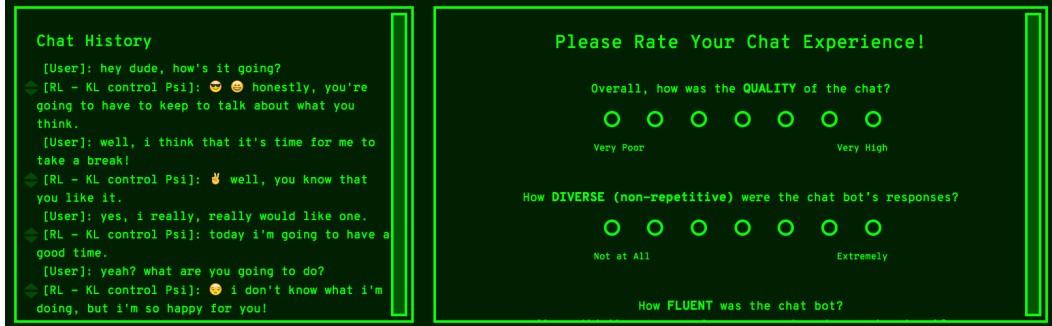

Figure 8: Interactive evaluation ratings page available at `https://neural.chat`.

Figure 9 is an example conversation within the platform that interactive evaluation participants see. Annotators can optionally click the up and down arrows beside each chatbot response to give feedback on the specific utterance. Once 3 or more turns of the conversation has taken place, participants may click "Close Chat and Rate" to get to the rating screen.

Figure 9: Interactive evaluation chat interface.

### B.1    WEBSITE SERVER SETUP AND CONFIGURATION

The server was hosted on a Google Cloud Platform virtual instance with 64GB of RAM and a NVIDIA Tesla P100 graphics card. The backend was a Django program being served by NGINX and uWSGI. For simplicity, we opted to have the Django process import the chatbots into the same Python process as Django, rather than have the two connect to each other via other means such as sockets. This configuration decreased development time and increased reliability, but it would need to be revisited if the server needed to scale several orders of magnitude past what was required for this study. The current configuration was still able to support hundreds of simultaneous users and host more than 30 bots concurrently.

The chatbots were kept in a separate project from the Django project and maintained separately from the server code. Each chatbot extended an abstract class that defined key methods for the Django program to use, and was registered to a globally accessible dictionary via a decorator. The Django project was provided the path to the Chatbots project in its PYTHONPATH, so it could import the dictionary in which all the chatbot objects had been registered and use that to dynamically determine which chatbots were available and to access them in its views.

It is important to note that the chatbots used PyCUDA, and PyCUDA does not work in a multiprocessing environment. Because of this, uWSGI needed to be configured to only have one python

process and to disable any attempt at multiprocessing. Furthermore, the chatbots required substantial startup times, so all chatbots are kept in memory at all times in the Django process. In order to keep all the chatbots in memory concurrently, we needed a very high amount of RAM on our server and opted for a 64GB virtual instance, and a GPU with 16GB RAM. This combination of CUDA to run the chatbots on the GPU with a high amount of RAM to keep all bots in memory at the same time resulted in incredibly fast server response times, with effectively no increase in response time when using the bots in requests compared to requests that did not.

For further information and instructions on server configuration, please read the server documentation available at *¡URL redacted for anonymity¿*. We hope that this platform will allow others to host their own bots and evaluate them in an interactive setting.

