# OpenReview forum: "Way Off-Policy Batch Deep Reinforcement Learning of Human Preferences in Dialog"
_ICLR.cc/2020/Conference — Reject_

### Official Review · AnonReviewer1 · 2019-10-23
**Official Blind Review #1**

**Rating:** 3

**Review:**

The paper introduces an off-policy batch deep reinforcement learning method for learning human preferences in dialog generation. The key technical contribution is by controlling the KL divergence of the learned policy with a prior policy that was learned on other dialogs. The method is able to constrain the policy not deviate too much from the prior to avoid extrapolation error. Another technical contribution is the use of dropout uncertainty estimates to estimate the Q values, which is more scalable compared to double Q-learning. The authors also incorporate intermediate implicit rewards in dialogs to encourage more positive conversations.
In terms of methodology, the use of dropout as uncertain estimates seem to be firstly used for DRL to my knowledge. The experiments seem to be properly conducted, with particular focus on inferring what rewards encourage positive conversations.

However, I am not fully convinced by the motivation of using KL control. If there is too much weight on the rewards of learning of human preferences compared to reasonable sentences, one could imagine other compared methods such as batch Q-learning could fail since they will learn to encourage positive conversations and diverge away from realistic language. Using KL to control its generated dialog to the prior is a way to prevent it from diverging. But I would think that with a proper tuning of the different type of rewards could also prevent the policy from diverging to generating non-realistic dialogs especially when there is a large amount of dialogs available. If possible I would hope to see tuning on the weights of different types of rewards used in the problem setup. Also I am curious if you use the same initialization (using the same pretrained language model) for other baselines? I did not seem to find it the paper.
Also the proposed method using KL control shares similarity to the one in Distral [1] where a distilled policy is used as prior policy and discounted KL divergence is used as regularizer, which limits it novelty in this perspective.

Overall, the paper is fairly well-written. The paper develops a method for learning human preferences in dialogs in off-policy reinforcement learning and use KL control to avoid extrapolation error issues. The technical novelty of the method is limited as there is a similar method proposed in [1]. The experiments are illustrative but more comparisons with the other methods will be appreciated to make it more convincing.

[1] Teh, Yee, Victor Bapst, Wojciech M. Czarnecki, John Quan, James Kirkpatrick, Raia Hadsell, Nicolas Heess, and Razvan Pascanu. "Distral: Robust multitask reinforcement learning." In Advances in Neural Information Processing Systems, pp. 4496-4506. 2017.

**Experience Assessment:**

I do not know much about this area.

**Review Assessment: Checking Correctness Of Derivations And Theory:**

I assessed the sensibility of the derivations and theory.

**Review Assessment: Checking Correctness Of Experiments:**

I assessed the sensibility of the experiments.

**Review Assessment: Thoroughness In Paper Reading:**

I read the paper at least twice and used my best judgement in assessing the paper.

---

> ### Author Response · Authors · 2019-11-15
> **Response to Review #1**
>
> We thank reviewer one for the feedback and evaluation.
>
> We respectfully disagree with the reviewer that solely maximizing the reward could prevent the policy from diverging to generate non-realistic dialog. One of the biggest challenges of a dialog task is that rewards that fully encapsulate conversation quality are difficult to generate and often easy to exploit. For example, a high reward text could be a long string of question marks and “haha”s. It is easy to sacrifice fluency and coherency for high reward. This is where using KL control with the learned prior is vital to maintain both realistic dialog and optimize for good dialog (high reward).
>
> The same initialization (same pre-trained language model) is used for all baselines (Table 1). Otherwise, there would not be enough training data to generate coherent sentences from the reward signals alone.
>
> In Section 2, we have extensively written about the previous work using KL-control to regularize divergence from a prior policy. We will expand the references to include [1] (on-policy setting). However, this work has major differences due to being completely off-policy where the policy is unable to explore and gather more data.

---

### Official Review · AnonReviewer2 · 2019-10-24
**Official Blind Review #2**

**Rating:** 3

**Review:**

This paper presents the Way Off-Policy (WOP) algorithm which penalize divergence fro a strong pre-trained prior model as a way to avoid extrapolation error and instability in batch reinforcement learning (BRL). KL-divergence penalties have previously been explored extensively in the on-policy setting (mainly with policy gradient methods) and the authors claim they are the first to apply this idea to off-policy RL. I am not familiar enough with related work to evaluate this statement, but given it’s true, this is an interesting application of these ideas. I also like the application to dialog because pre-trained language models have been shown to offer very good priors for high-quality natural language generation.

Unfortunately, before any other comments I would like to point out that this paper is not properly anonymized. Both the submitted code and the online demo that is linked to from the paper contain the author names and their affiliation (the page linked to in Section 1 contains the author names but the other one in Section 5 does not—it mentions they have been redacted). Apparently there is also an arXiv version of the paper which I was not aware of and which would be fine if this version was not cited from the README file in the submitted code. Please be more careful with respect to anonymizing your submission!

Overall the paper proposes an interesting algorithm that combines existing ideas in a new way and arrives at an algorithm that achieves good performance, based on the experimental evaluation provided in the paper. The experiments performed on dialog are good and I really like that human evaluation was used. However, the traditional RL experiments are lacking. The authors only present results for the CartPole environment in the main paper, which is a very easy problem and so not that interesting. In the appendix they also show results for the Acrobot environment, which are a bit weaker for WOP. Both are quite simple problems and it’s quite easy to learn a good prior model. Why not present results for harder problems? It should be easy to run on more Gym environments and see if WOP still does well in harder problems. Other than that, I like the main ideas in the paper, but I think they could be presented better. Given this and the anonymization issue I lean towards rejecting this paper for now.

Other comments:

(1) I don’t necessarily agree with the statement “Because learning language online from humans on the internet can result in inappropriate behavior (see Horton (2016)), learning offline using BRL is imperative.” I would rather just say that learning offline is one way of dealing with this issue but it’s not the only one or necessarily the best one. In general, the paper defends the batch RL setting quite extensively, but I feel some of the statements may be overly strong. There is a lot of prior work that shows when and why the batch RL setting is useful and so I would add more references to such work.

(2) In Section A.1 you present the implicit metrics that were used and the weight of each metric. These weights seem *extremely* arbitrary to me so could you please provide some insight (and references) as to how they were chosen? I like the results presented in Table 3, in terms of the insight they provide. Are these what you used to derive the weights presented in Section A.1?

(3) The yellow shading in Figure 1 is not really visible. You could try using a different color for the shading (e.g., use gray and make the walls blue or something like that).

(4) The axis labels in the plots of Figure 2 are too tiny. These plots should be made bigger to be readable in printed form. Also, I would recommend you order the labels in the legend in maybe chronological order or something like, which would also put WOP at the top to clarify it’s the proposed method.

Disclaimer: I am not too familiar with this area of research and the related work and so my comments and evaluation should be taken into account in that context.

**Experience Assessment:**

I have read many papers in this area.

**Review Assessment: Checking Correctness Of Derivations And Theory:**

I did not assess the derivations or theory.

**Review Assessment: Checking Correctness Of Experiments:**

I assessed the sensibility of the experiments.

**Review Assessment: Thoroughness In Paper Reading:**

I read the paper thoroughly.

---

> ### Author Response · Authors · 2019-11-15
> **Response to Review #2**
>
> We like to thank reviewer two for the thoughtful evaluation of not only our paper but our supplementary materials as well. We do apologize for the incorrect link that contained deanonymized information. We will take this learning and be more careful in future submissions.
>
> We will add more experiments of our off-policy algorithm. For this contribution, we are motivated by the practical and unsolved problem of how RL can be applied to Dialog problems through our algorithm. We include CartPole and Acrobot to demonstrate that this WOP is not only applicable in Dialog.
>
> In Section A.1. the coefficients for the implicit metrics come from a linear fit of the implicit metrics to actual human ratings. We do this to translate human evaluation in the form of final scores provided into features of the conversation that can be evaluated at each step. We will update the supplementary materials with additional details to clarify how these weights have been estimated.

---

### Official Review · AnonReviewer3 · 2019-10-24
**Official Blind Review #3**

**Rating:** 3

**Review:**

This paper proposes an off-policy batch reinforcement learning algorithm, Way Off-Policy(WOP). The authors address the extrapolation error which is a general problem of batch reinforcement learning. WOP algorithm uses KL-control to penalize divergence between prior and policy. Detailed comments are as follows:

- KL penalize is the first in batch RL, but many approaches have already been proposed in RL. Moreover, I'm not sure if it is novel because I think that it is not much different from KL penalize proposed in RL.
- And DBCQ seems to simply remove the perturbation of BCQ. Is there any other difference? (If so, a detailed explanation needs to be added in the paper.)
- Are there any comparisons with more recent batch RL algorithms for discrete actions? (ex. SPIBB(Safe Policy Improvement with Baseline Bootstrapping, Laroche R., et al 2019))
- Although the main experiment is about dialog, the section of ‘traditional RL experiments’ seems not enough for cartpole only.

This paper is well organized and clearly written, but I cannot agree with the main contribution (KL-control penalize) of this paper (because the KL penalize idea is very similar to proposed approaches in RL). And, it seems to be compared with more recent batch RL algorithms in the experiments.

[Minor errors]
- c first appears in Equation (6), which seems to have no definition and explanation in the paper.
- It is shown that the normalization constant is missing in Equation (9).
- In section 3.4, line 4, pi_\theta(a_t, s_t) -> pi_\theta(a_t | s_t)


**Experience Assessment:**

I do not know much about this area.

**Review Assessment: Checking Correctness Of Derivations And Theory:**

I assessed the sensibility of the derivations and theory.

**Review Assessment: Checking Correctness Of Experiments:**

I assessed the sensibility of the experiments.

**Review Assessment: Thoroughness In Paper Reading:**

I read the paper at least twice and used my best judgement in assessing the paper.

---

> ### Author Response · Authors · 2019-11-15
> **Response to Review #3**
>
> We would like to thank reviewer three for the suggestion to compare against batch RL algorithms for discrete actions. For a fair comparison, we chose BCQ due to highest conceptual similarity to our technique and extended it to discrete space (DBCQ). We look forward to adding additional batch RL baselines.
>
> In traditional RL experiments, we also demonstrate WOP on Acrobot in the supplementary materials (Section A.3.1)

---

### Decision · Program_Chairs · 2019-12-19

**Decision:**

Reject

**Comment:**

This paper offers a possibly novel approach to regularizing policy learning to make it suitable for large-scale divergence in the underlying domain.  Unfortunately all the reviewers are unanimous that the paper is not acceptable in present form.  Insufficient clarity regarding the contribution relative to several references, some of which were missing from the submitted version, is perhaps the most significant issue in the view of the AC.